# Supporting evidence-based decisions about the geographic and demographic extensions of seasonal malaria chemoprevention in Benin: A modelling study

Jeanne Lemant[1,2], Clara Champagne[1,2,3], William Houndjo[4], Julien Aïssan[4], Rock Aïkpon[4], Camille Houetohossou[4], Sakariahou Kpanou[4], Roland Goers[1,2], Cyriaque Affoukou[4], Emilie Pothin[1,2,3]*

1 Swiss Tropical and Public Health Institute, Allschwil, Switzerland, 2 University of Basel, Basel, Switzerland, 3 Clinton Health Access Initiative, Boston, Massachusetts, United States of America, 4 National Malaria Control Programme, Ministry of Health, Cotonou, Benin

* emilie.pothin@swisstph.ch

## Abstract

Seasonal malaria chemoprevention (SMC) has been implemented yearly in northern Benin since 2019 to reduce the malaria burden in children under 5 years of age. Its geographic scope was progressively extended until in 2022 two different extensions of SMC were considered: either demographic - children aged 5–10 in the currently targeted departments would also receive SMC, or geographic to children under 5 in new eligible departments to the south. As SMC had neither been implemented in the areas nor in the age groups suggested for expansion, modelling was used to compare the likely impact of both extensions. The model OpenMalaria was calibrated to represent the history of malaria interventions and transmission risk in administrative units of Benin. Currently planned future interventions and two scenarios for SMC extensions were simulated to inform where impact would be the highest. The model predicted that between 2024 and 2026 the geographic extension of SMC would avert at least four times more severe malaria cases and five times more direct malaria deaths per targeted child than the demographic extension. Indeed, most severe cases are concentrated in children under 5 in all departments of interest, as malaria burden remains high in this region. Numbers of severe cases averted per targeted child were similar between health zones eligible for geographic extension. The main limitations of this work are global model parameters due to lack of country-specific data on efficacy of interventions or development of immunity. SMC coverage was assumed to be uniform across rounds, zones, and age groups. Due to the high malaria burden in northern and central Benin, the geographic extension would be more impactful than the demographic extension both in absolute number of severe cases averted and per child protected, and has started to be implemented in 2024. Health zones were prioritised by availability of community health workers to deliver

**Data availability statement:** Model code can be found here: https://github.com/SwissTPH/openmalaria and here: https://github.com/SwissTPH/r-openMalariaUtilities. Code for data manipulation, input parameters and processing is available here: https://github.com/SwissTPH/BeninSMC

**Funding:** This work was supported by the Gates Foundation (INV-030449 to EP). Under the grant conditions of the Foundation, a Creative Commons Attribution 4.0 Generic License has already been assigned to the Author Accepted Manuscript version that might arise from this submission. The funder had no role in study design, data collection and analysis, decision to publish, or preparation of the manuscript.

**Competing interests:** The authors have declared that no competing interests exist.

SMC. Mathematical modelling was a supportive tool to understand the relative impact of the different proposed SMC extensions and contributed to the decision-making process. Its integration significantly enhanced the utilisation of data for decision-making purposes. Rather than being used for forecasting, the model provided qualitative guidance that complemented other types of evidence.

## Introduction

Malaria killed 597 000 persons worldwide in 2023, of which 73.7% were children under 5 [1]. This parasitic disease is one of the leading cause of death in children under 5 [2]. Strong reductions in malaria burden have been achieved since 2000, mainly thanks to insecticide-treated nets (ITNs) and artemisinin-based combinations therapies [3], but since 2015 progress has stalled. Insecticide and drug resistance, invasion of new vector species, and lack of funding threaten to reverse the decreasing trend. Additional interventions to nets and treatments are required to further decrease transmission and protect those most at risk of severe malaria. Seasonal malaria chemoprevention (SMC) serves exactly this purpose, consisting of administering antimalarials to children living in seasonal transmission areas during the period of heightened malaria risk to clear existing infections and prevent new ones [4]. The number of children treated increased from 0.2 million in 2012 when the World Health Organisation (WHO) recommended SMC to 49 million ten years later. In 2022 the WHO updated their recommendations for SMC: the intervention is no longer recommended only to children under 5 years of age, but to all children at high risk of severe malaria, without specific age groups or transmission intensity thresholds [4,5]. The increased flexibility is highlighted by the WHO as an opportunity for malaria endemic countries to better tailor their strategies to local needs, and National Malaria Control Programmes (NMCPs) may explore the possibility to expand the intervention to new areas (geographic extension) or to older children (demographic extension) [5].

SMC is an operationally feasible intervention which can reach coverages above 90% [6,7]. The intervention has been shown to be efficacious in numerous randomised controlled trials in children under 5 and under 10. Cissé et al. [8] administered SMC to children under 10 and observed incidence in children under 5 decrease by 57% and in children between 5 and 9 by 61% during the transmission season, while others found even larger reductions [9–11]. To our knowledge, no study has compared the benefits of extending the age limit of SMC to those of expanding to new zones, but NMCPs need to know which strategy is more effective in their context to make an informed decision.

Most severe malaria cases and deaths occur among children under 5 especially under high transmission intensities [12–14], which calls for a geographic extension of SMC to protect those most at risk. But older children can represent a large parasitic reservoir [15] and as young children are protected, burden may shift to school-aged children [16]. Several countries have already adopted interventions targeting school-aged children, such as Intermittent Preventive Treatment in Tanzania [17] or SMC in children under 10 in Mali [18]. Moreover, the impact of SMC in different age groups

may be influenced by factors such as intensity of transmission, seasonality or the concomitant deployment of other interventions, which vary both across and within countries. Therefore, it is not straightforward to anticipate which SMC strategy would be most impactful in a given context, and setting-specific analyses using the available local data are required, as encouraged by the WHO [5,19].

In Benin, malaria is the first cause of consultation and accounts for 2% of global malaria cases and 1.7% of deaths [1]. SMC was first implemented in 2019 in the health zones of Malanville-Karimama (Alibori) and Tanguieta-Materi-Cobly (Atacora). The geographic scope of SMC was progressively extended until in 2021 almost 600 000 children under 5 living in the two Sahelian departments of Alibori and Atacora were targeted (NMCP, personal communication). After the update of the WHO recommendations, the Benin NCMP considered either extending SMC to children under 10 years of age in Alibori and Atacora, where SMC was already implemented, or extending SMC to new eligible zones in the departments of Borgou, Collines, and Donga but only targeting children under 5. Choosing between these two options was necessary as the entire population of Benin is at risk for malaria, but funding is limited. This also constrained the number of new zones which could be targeted, so the NMCP was interested in a ranking of the new eligible zones based on impact to inform the prioritisation within the targeted departments.

Beyond determining which of the two extension strategies would save more lives, understanding their relative impact would enable the NMCP to weigh this likely impact against the required implementation efforts. Such a quantification will intrinsically be context-specific and as such requires a method which can be adapted to the setting of interest. Mathematical modelling can mimic the past and current malaria burden in administrative units of a country and simulate the impact of various interventions scenarios to identify the most promising ones. It has already been used to support decision-making at the global level [20,21] and within countries [22–24]. However modelling outputs are often used as advocacy tools [25] and only rarely taken up by countries' health authorities to guide malaria national policies [22].

In this work, we illustrate how we used mathematical modelling to inform the new SMC policy in Benin. First, we adapted a pre-existing individual-based model of malaria transmission to reproduce malaria dynamics in each of Benin's 77 communes. Second, we quantified the impact of both SMC extension strategies, demographic and geographic. Finally, we provided a ranking of the new eligible zones according to their predicted SMC impact.

## Methods

We used the OpenMalaria modelling suite [26,27] to simulate the impact of interventions on malaria burden in Benin. OpenMalaria comprises an agent-based model of human malaria infections including immunity and superinfection [26], together with a compartmental model to govern the evolution of the *Anopheles* vector population [28] and it has been described extensively elsewhere [29,30]. Simulations were set up and run using the R package OpenMalariaUtilities [31].

The model was adapted to Benin to make predictions tailored to the context of the country. The entire workflow is summarised in Fig 1.

### Model parameterisation: capturing historical trends for each commune up to 2023

Benin is divided into 12 departments (first administrative level) and 77 communes (second administrative level). The operational level consists of health zones, which comprise one or several communes. The model was simulated for each commune in the country. We collated global data on interventions efficacy and local data on past deployments in Benin and adjusted the modelled transmission risk in each commune so that yearly modelled prevalence would match historical prevalence trends. Model parameters are summarised in Table 1 and details of assumptions by department and communes are in S1 Appendix.

**Benin-specific parameterisation of OpenMalaria.** The dominant vector species in Benin is *Anopheles gambiae sensu lato* [32], with *Anopheles funestus* as a secondary species for which a parameterisation was available; in the model they were considered to contribute to transmission at 90% and 10% respectively. We used data on human sleeping

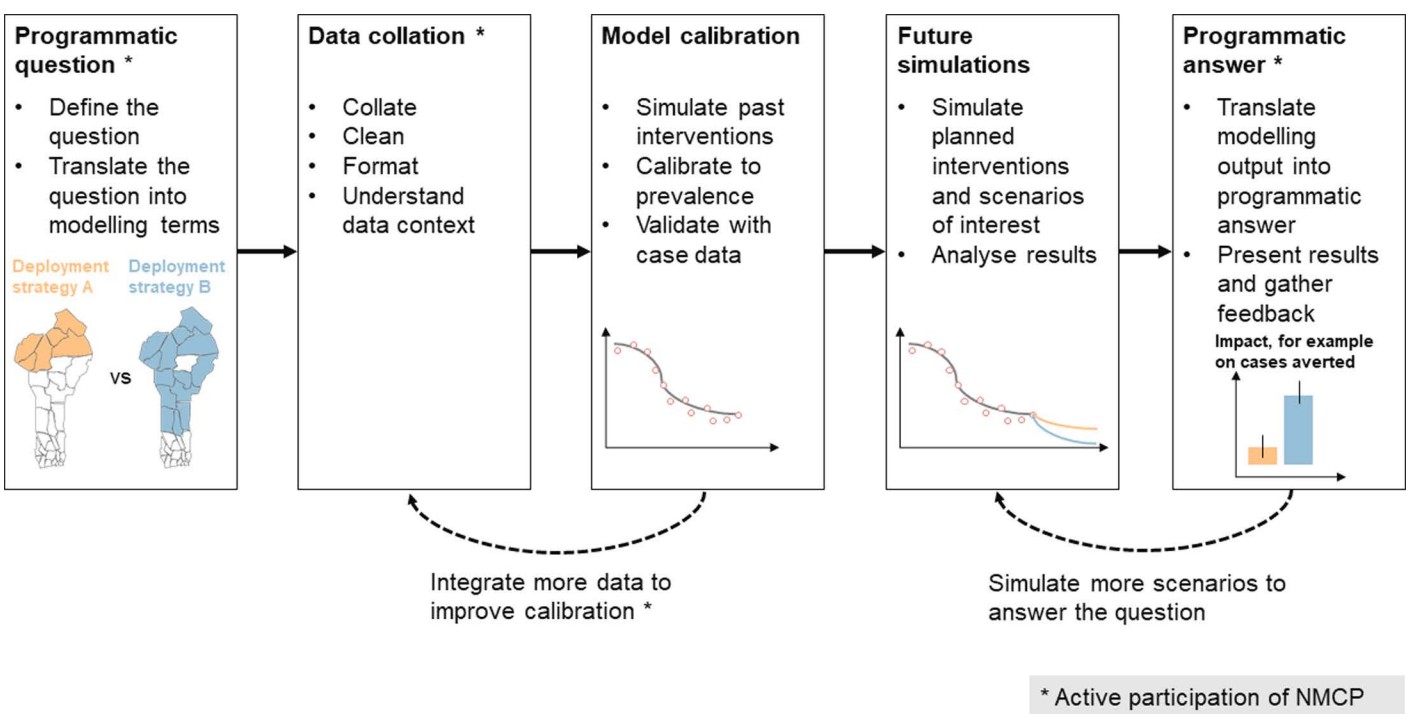

**Fig 1. Applied workflow to answer programmatic questions with country-specific mathematical modelling.** Schematics are for illustrative purpose only. Base layer of Benin map from https://data.humdata.org/dataset/cod-ab-ben.

patterns and mosquito biting rhythms [33] to compute the exposure of humans to bites outdoors, where mosquito nets and insecticide spraying are ineffective. Intense pyrethroid resistance was first recorded by sentinel sites in 2016 (NMCP, personal communication) so from 2016 vectors were assumed to be fully resistant to pyrethroids, with nets acting only as a physical barrier. Before that, we assumed vectors were moderately resistant to pyrethroids, using a previously published parameterisation of ITNs [36].

Seasonality of transmission was informed by reported incidence in each department [34], with a 15-day lag to account for the time between infection and onset of symptoms.

Before 2000, the model is assumed to be at equilibrium. We then included the history of malaria control interventions deployed between 2000 and 2019, namely vector control, case management and SMC. Since household surveys are powered at the department level, history of interventions was simulated at the department level except when information was available at the lower commune level, such as for insecticide residual spraying (IRS) and SMC.

We used ITN use estimates from the Malaria Atlas Project [37] at the department level until the first mass distribution campaign in 2011. We then relied on household surveys [38,39] to estimate ITN use, assuming nets distributed during each mass campaign had a three-year half-life [40]. Distributed ITNs were pyrethroid-only nets until 2019, with piperonyl butoxide (PBO) introduced in 2020, and chlorfenapyr-pyrethroid nets in 2023. Both these next-generation nets were modelled as restoring most susceptibility in vectors, based on an experimental hut trial recording 70% mortality in bioassays [36]. Durability based on net fabric was also taken into account, with nets made of polyester or polyethylene having a half-life of two - respectively 1.6 – years [41]. We made the conservative assumption that for all distributions after 2017, ITN use just after distribution was similar to that measured in the last available survey in 2017. IRS active ingredient, population coverage and month of spraying were modelled at the commune level until the intervention was discontinued in 2022 [42]. Insecticide efficacy parameterisations were derived from experimental hut trials results [33,43].

**Table 1. Model parameters and data sources.**

| Parameter | Value | Source |
|---|---|---|
| **Vector species contribution** | *Anopheles gambiae sensu lato* 90%<br>*Anopheles funestus* 10% | [32] |
| **Proportion of bites occurring outdoors** | 20% for *Anopheles gambiae sensu lato*<br>10% for *Anopheles funestus* | Computed from human sleeping patterns and mosquito rhythms from Benin and neighbouring countries using the AnophelesModel R package [33] |
| **Seasonality of transmission** | Based on monthly incidence by department, averaged between 2011 and 2017 with a 15-day lag | Cases from national surveillance [34], population from census [35] |
| **Types of nets** | Only pyrethroid-only nets until 2020<br>Mixture of pyrethroid-only nets and PBO in 2020, and of pyrethroid-only nets, PBO and chlorfenapyr nets in 2023 | Matching |
| **Efficacy of nets** | From experimental hut trials<br>From 2016, pyrethroid-only nets are modelled as a physical barrier only, to account for widespread insecticide resistance | [36]<br>PBO and chlorfenapyr nets are both modelled as restoring susceptibility to insecticide |
| **Net use** | Specific values per year and per department.<br>National average value: 6.5% in 2000, 71.1% in 2017 | 2000 – 2010: MAP [37]<br>2011 – 2023: based on latest household survey at distribution time [38,39] |
| **Net durability** | Before 2020: 3-year half-life<br>After 2020: Depending on fabric: 2-year halflife for polyester nets (pyrethroid-only nets, Interceptor G2 and PermaNet 3.0), 1.6 for polyethylene (DuraNet) | Before 2020 [40]<br>After 2020 [41] |
| **IRS implementation** | IRS active ingredient, population coverage and month of spraying estimated from PMI Malaria Operational Plans<br>Coverages between communes and years ranged from 47% to 90% | [42] |
| **IRS efficacy** | From experimental hut trials | [33,43] |
| **Effective treatment coverage** | Uncomplicated cases: specific values per year and per department<br>National average value: 22.8% in 2001, 28.1% in 2018<br>Severe cases: 48% | Uncomplicated cases: methodology by [44], with data from [38,39]<br>Severe cases [45] |
| **SMC efficacy** | 25 days blood clearance | Calibrated to match effect size in [46] |
| **SMC coverage** | 80% of children at random at each of the four rounds from July to October | Reported coverage from the 2021 campaign |
| **Age structure** | Benin census | [35] |
| **Importation rate** | 10 infections per 1000 people per year in each commune | [47] |
| **Parasite detection limit** | 100 parasites per microliter | [48] |
| **Malaria prevalence** | Geospatial estimates of prevalence in children aged 2–10, based on household surveys in children under 5<br>National prevalence in children under 5 in Benin in 2017: 36.4%<br>Alibori: 36%<br>Atacora: 53.9%<br>Borgou: 51.3%<br>Collines: 41.6%<br>Donga: 40.2% | [38,49] |

Access to treatment in case of fever has been reported in household surveys between 2001 and 2017 [38,39] at the department level. Once individuals seek treatment, they do not necessarily receive a completely curative treatment due to care-seeking behaviour in unofficial structures, lack of compliance of care providers, lack of adherence of certain patients, presence of counterfeit medicines and imperfect efficacy of treatments. The effective treatment coverage used in the model is therefore the product of the access to health services measured during each of the five surveys by those factors which quantify the inefficiencies in the care management cascade [44]. After 2017 effective treatment coverage was assumed to remain constant. While access to treatment for uncomplicated malaria cases was derived from household surveys, for severe malaria cases we assumed that 48% of patients seek care [45].

In Benin an SMC cycle consists of administering SP-AQ (sulfadoxine-pyrimethamine and amodiaquine) monthly during four rounds from July to October. After Malanville-Karimama (Alibori) and Tanguieta-Materi-Cobly (Atacora) in 2019, SMC was extended to Banikoara and Kandi-Gogounou-Segbana (Alibori) in 2020, and finally all children under 5 of both departments were targeted from 2021. Efficacy of SMC was modelled through a blood-clearance effect lasting 25 days to reproduce the effect size observed in a trial in the Sahel [46]. This means SMC is assumed to clear all infections for 25 days, then its effect wanes instantly. To match the lowest reported coverage from the 2021 SMC campaign, 80% of eligible children under 5 were randomly selected to receive SMC at each of the four rounds. This means each child was equally likely to receive SMC at each round.

The model reproduces the population age structure from the Benin 2013 census [35]. We also assumed 10 infections per 1000 people per year were imported in each commune, a conservative assumption [47]. Parasite detection limit was set at 100 parasites per microliter to reproduce rapid diagnostic test detection limit [48]. Since we only needed prevalence for the calibration and no rarer events such as severe cases or deaths, we simulated a small population of 3,000 individuals. This was also a trade-off to avoid running many long simulations.

**Model calibration.** The transmission level, represented by the pre-intervention Entomological Inoculation Rate (EIR) in 2000, was adjusted for each commune to match the time series of past prevalence estimates for years 2006–2019 provided by the Malaria Atlas Project (MAP) [49].

For each department, we computed a grid of simulations sharing the same history of interventions and geographical characteristics but with EIR values ranging from 1 to 250 (with a step of 1 until 10, and a step of 2 afterwards), and 10 stochastic replicates. The associated OpenMalaria simulations for prevalence over the years 2006–2019 are noted $X^{sim,\ \theta} = \left(X_i^{sim,\theta}\right)_{i \in \{2006,\dots,2019\}}$, with $\theta$ denoting each simulation's unique set of parameters (including the initial EIR).

MAP's prevalence estimates [49] are noted $X^{obs} = \left(X_i^{obs}\right)_{i \in \{2006,\dots,2019\}}$ and their associated 95%-confidence intervals $(X_i^{low}, X_i^{up})$ were used to compute an estimated standard deviation $\sigma = (\sigma_i)_{i \in \{2006,\dots,2019\}}$ as $\sigma_i = \frac{\max(X_i^{obs} - X_i^{low}, X_i^{obs} + X_i^{up})}{1.96}$. For each simulation in the grid, a pseudo-likelihood comparing model prevalence to MAP prevalence is computed as

$$L\left(X^{obs},\ X^{sim,\ \theta}\right) = \Pi_{i=2006}^{2019} \Phi_{X_i^{sim,\ \theta}, \sigma_i}\left(X_i^{obs}\right)$$

where $\Phi_{\mu,,\sigma}$ represents the probability density function of a normal distribution with mean $\mu$ and standard deviation $\sigma$. The EIR value with highest log-likelihood ($E^{point}$) is selected as point estimate.

An uncertainty estimate is computed using the profile likelihood approach by Ionides et al. [50] adapted to this particular use case. In Ionides et al. [50], a local quadratic approximation is fitted on the smoothed pseudo-likelihood and used to calculate the threshold used in the definition of the confidence intervals. For each commune, the local window for the quadratic approximation was defined by the EIR values for which $\frac{|E^{point} - EIR|}{E^{point}} \leq 0.5$ and the quality of the quadratic approximation was assessed graphically. Thanks to this method, we obtained upper and lower uncertainty bounds on the initial EIR ($E^{low}$ and $E^{up}$) for each commune.

In the future scenarios, for each of Benin's 77 communes, OpenMalaria was run for $E^{point}$, $E^{low}$ and $E^{up}$, in order to propagate the uncertainty in transmission intensity due to prevalence data fitting. Each future simulation also includes 10 replicates to account for model stochasticity. The final uncertainty intervals represent the ensemble of both sources of uncertainty.

## Simulation of SMC extensions after 2024

The future scenarios were simulated from 2024 to 2026, which was the period of interest defined by the NMCP as the funding period. Some interventions were already scheduled by the NMCP, such as continuing SMC campaigns for children under 5 in Alibori and Atacora and distributing next-generation nets in 2026 in the whole country. The Plus Project, a pilot study of perennial malaria chemoprevention (PMC) targeting children until 2 years, had also started in some

communes (Fig 2). These interventions, although not yet fully deployed when the modelling work was conducted, were confirmed to happen and needed to be included in the model for a comprehensive representation of malaria control over the period 2024–2026. Additionally to these already planned interventions, we simulated both SMC extension scenarios starting in July 2024. The administration period, coverage, and efficacy were assumed to be identical to the already planned SMC campaigns. For the demographic extension of SMC we simulated its administration to children from 5 to 10 in Alibori and Atacora. For the geographic extension the NMCP had already identified the eligible zones situated in the region of the Sahel and where more than half the cases were occurring within four consecutive months. We simulated the administration of SMC to children under 5 in Borgou, Collines, and Donga except in the health zone of Bembereke-Sinende (Borgou), since SMC and PMC are mutually exclusive and this zone was already part of the PMC pilot project.

In order to measure severe cases and deaths which are rare events, we used a simulated population size of 50,000 individuals. This was the population size of the least populated commune in 2024, Toucountouna, to avoid simulating a population larger than the actual population (which would have artificially reduced the demographic stochasticity and underestimated the uncertainty), and already resulting in a runtime of up to two hours per individual simulation. All epidemiological indicators were then rescaled to the real population size in each commune for each year, calculated by applying a 3.51% yearly growth rate (as measured nationally between 2002 and 2013) to the population of each commune measured during the 2013 census [35].

## Comparison of extension scenarios

The indicators of interest were the total number of malaria episodes, severe cases, and deaths directly caused by malaria. Malaria episodes count clinical malaria infections, regardless of treatment or reporting. Any infection which lasts more than 30 days counts as several episodes to mimic the health system memory [51]. We computed epidemiological indicators $II$ (episodes, severe cases and deaths) averted by each extension scenario by comparing those occurring with only the planned interventions to those with either the demographic or the geographic extension of SMC. Each simulation is uniquely defined by a commune, a stochastic replicate $s \in \{1, \ldots, 10\}$, an EIR value $E \in \{E^{low}, E^{point}, E^{up}\}$ and a scenario. We computed the mean averted indicator $A^{point}_{extension}$ between the planned interventions and an extension scenario as such:

$$A^{point}_{extension} = \underset{s \in \{1,\ldots,10\}}{mean} \left( I^{s,E^{point}}_{planned} - I^{s,E^{point}}_{extension} \right)$$

and the lower bound of the credible interval

$$A^{low}_{extension} = \underset{\substack{s \in \{1, \ldots, 10\} \\ E \in \{E^{low}, E^{point}, E^{up}\}}}{min} \left( I^{s,E}_{planned} - I^{s,E}_{extension} \right)$$

(respectively taking the maximum for the upper bound). We computed the extremum over the EIR as well as the stochastic replicates since for close EIR values, a lower EIR can produce a slightly higher indicator.

We summed each averted indicator between 2024 and 2026 over all communes of Alibori, Atacora, Borgou (except Bembereke-Sinende), Collines and Donga.

To account for different population sizes between zones, we also divided the episodes, severe cases and deaths averted by the newly targeted population. For the demographic extension we divided by the number of children between 5 and 10 in Alibori and Atacora, and for the geographic extension by the number of children under 5 in the eligible health zones of Borgou, Collines, and Donga. This way, even if the population growth were stronger than planned, we were comparing indicators averted by additionally targeted population instead of absolute numbers. The number of newly targeted children is also used as a measure of the effort of extending SMC.

## Results

### Reproducing Benin's malaria trends at the commune level

In each of Benin's 77 communes, we calibrated the OpenMalaria model to reproduce the trend in annual malaria prevalence between 2006 and 2019, as estimated by the Malaria Atlas project [49]: the resulting modelled prevalence at the national level is displayed in Fig 3A (commune-level results are presented in S1 Appendix). The model captures the overall prevalence trend, which decreased from 44% (41, 46) in 2006 to 31% (28, 34) in 2020, although a rebound in the years 2010 was observed (Fig 3A). This rebound happens later in our model predictions compared to MAP's estimates, leading to a difference in trends between 2012 and 2017, but overall, uncertainty intervals overlap for all years except 2015 and 2016. MAP's yearly prevalence estimates are based on household surveys which are conducted during a few months and are powered to be representative at the department level [38]. When comparing modelled prevalence in each department during the same months to those surveys, we obtained an adjusted $R^2$ of 0.79 (Fig 3B, 2011 and 2017 surveys).

Predictions of the trends in malaria cases and incidence between 2000 and 2023 are displayed in Fig 4. The total number of predicted malaria episodes increased from 8.1 (8, 8.3) million in 2000 to 17.8 (17.5, 18) million in 2023. This trend is largely driven by population growth, as the population in Benin more than doubled from 6.4 million inhabitants in 2000 to 14.1 million in 2023 [35]. Modelled malaria incidence in children under 5, which is not affected by population growth, decreased from 3392 (3269, 3466) malaria episodes per 1000 children in 2000, to 2378 (2196, 2492) in 2023, with a strong decrease after 2010 to reach its lowest value in 2016 and a re-increase since. A similar trend, although more moderate, was observed in all-age incidence. The strong incidence reductions after 2010 can be attributed in the model to the large scale deployment of malaria interventions, especially ITNs. The re-increase is due to the reduction in ITN efficacy

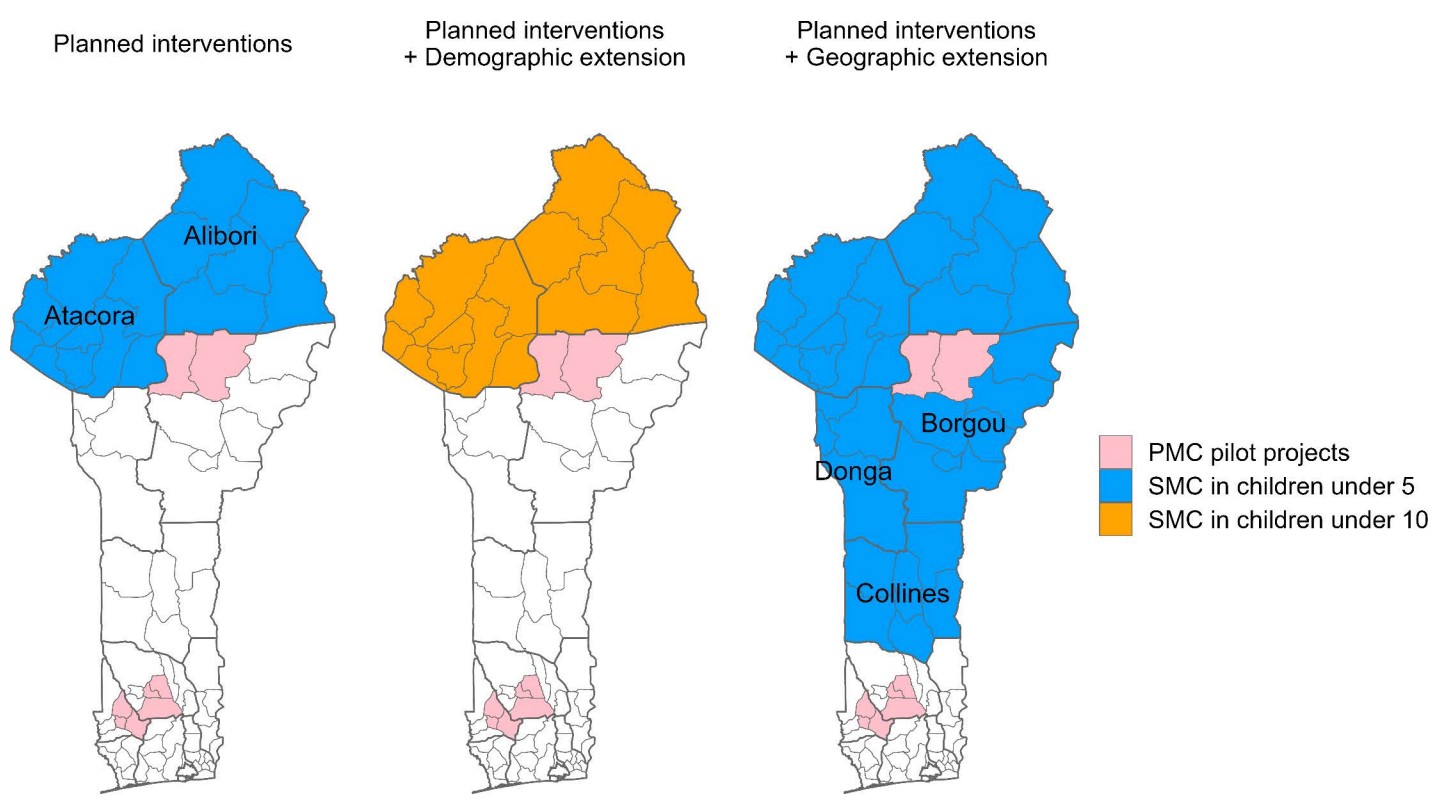

**Fig 2. Future scenarios for the period 2024 to 2026.**

due to insecticide resistance in 2016 and from the reduced acquired immunity in populations protected by ITNs over long periods of time [52].

Since the model was calibrated to prevalence, we could validate predictions of malaria cases to other sources, such as the World Malaria Report (WMR) 2024 [1] and routine surveillance data [34]. Modelled treated cases are infections which were detected and treated by the health system according to the effective treatment assumptions. For 2023, the model predicted 4.5 (4.4, 4.7) million treated cases, whereas the World Malaria Report estimated 5.1 (3.2, 7.7) million cases [1]. Modelled treated cases account for the inefficiencies in the case management cascade, but nationally reported cases are even lower since systematic testing and reporting of cases is still being improved in Benin.

From now on we only focus on the northern zones considered for the demographic (Alibori and Atacora departments) or geographic (Borgou except for Bembereke-Sinende where the pilot project of PMC is taking place, Collines and Donga) SMC extensions.

## Greater impact of geographic over demographic extension

The model predicted that 668 (573, 783) thousand malaria episodes in all age groups could be averted between 2024 and 2026 by the demographic extension of SMC (Fig 5), among these 3 (-1, 7) thousand severe cases. Additionally, 25 (-5, 57) direct malaria deaths in all age groups could be averted. By contrast, 1.5 (1.4, 1.5) million malaria episodes could be averted by the geographic extension between 2024 and 2026, among these 29 (22, 34) thousand severe cases. 244 (192, 289) malaria deaths would also be averted.

Between 2024 and 2026 we estimated there would be one million children between 5 and 10 years old in Alibori and Atacora to whom SMC would be additionally administered. In eligible zones of Borgou, Collines, and Donga, 1.8 million children under 5 would be eligible to SMC. When dividing cases averted by the additionally targeted population, we obtained that the demographic extension would avert 655 (562, 768) malaria episodes in all age groups per 1000

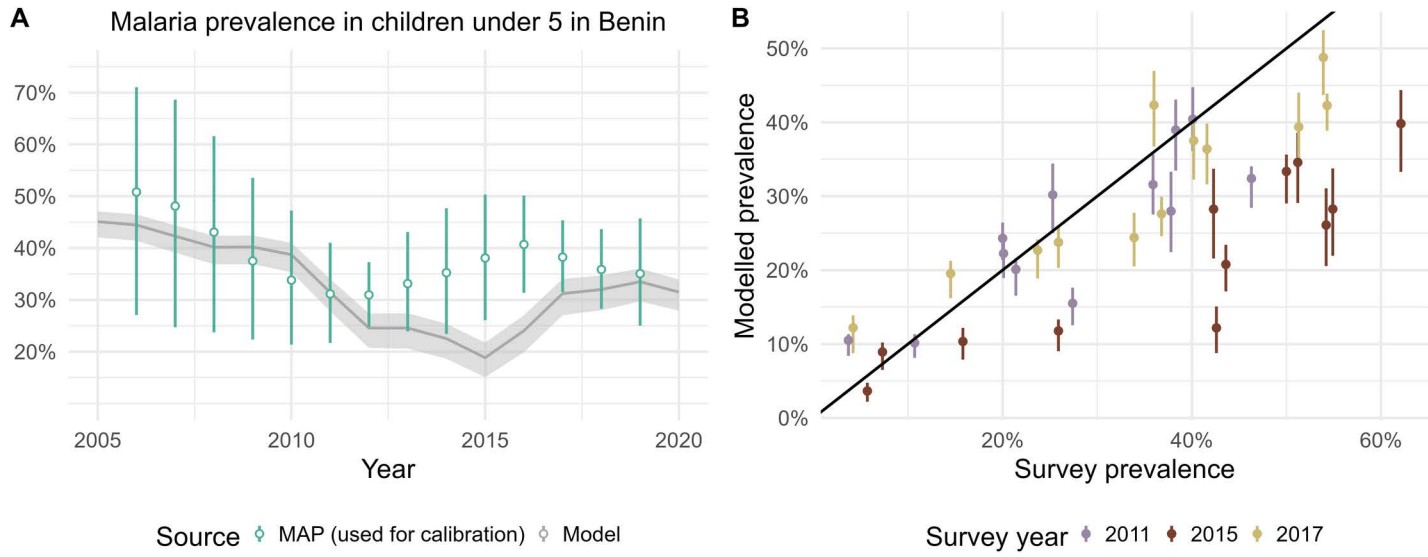

**Fig 3. Comparison of observed, estimated and modelled malaria prevalence in children under 5.** (A) Estimated (green, Malaria Atlas Project [49]) and modelled (grey) yearly malaria prevalence in children under 5 aggregated at the national level. (B) Modelled malaria prevalence in children under 5 against prevalence measured during household surveys in each department [38, 39], matched by survey period. The 2011 and 2017 surveys [38] were used to produce the MAP prevalence estimtes. Ribbon (A) and error bars (B) represent uncertainty on intensity of transmission as well as model stochasticity, as described in the Methods section.

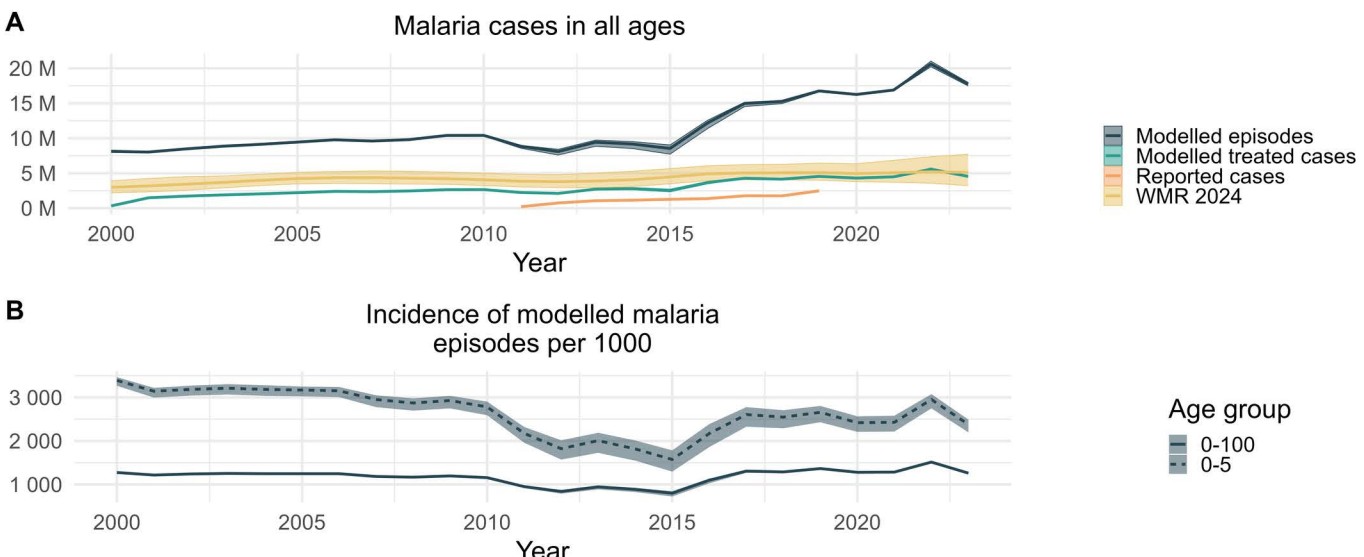

**Fig 4. Trends in absolute numbers (A) and incidence (B) of modelled case indicators.** (A) Absolute numbers of modelled malaria episodes and treated cases in all ages, as well as cases reported by Benin and estimates from the World Malaria Report (WMR) 2024 [1]. (B) Incidence of modelled malaria episodes per 1000 persons at risk in all age groups and children under 5. Ribbons represent uncertainty on intensity of transmission as well as model stochasticity, as described in the Methods section.

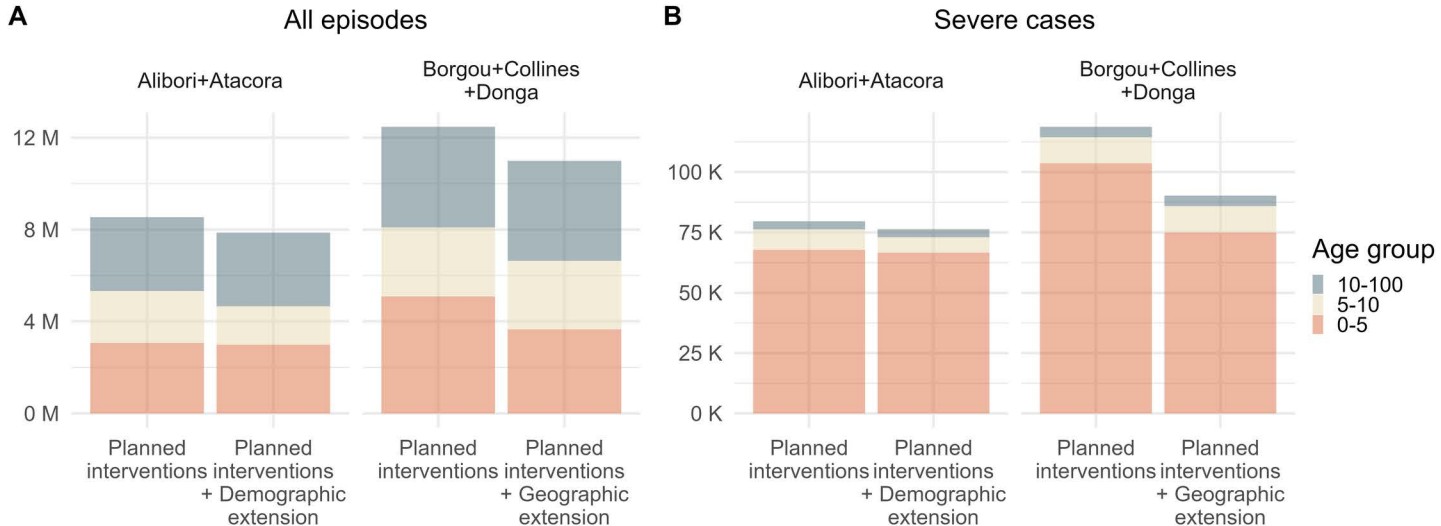

**Fig 5. Cases by scenario and age group between 2024 and 2026. Distribution of mean predicted malaria episodes (A) and severe cases (B) across age groups with planned interventions and adding SMC extensions in eligible zones. Bembereke-Sinende health zone has been excluded from Borgou since it is not eligible to the SMC extensions.**

targeted children from 5 to 10, and among these 3 (-1, 7) severe cases (Fig 6). The demographic extension would also avert 24 (-94, 56) malaria deaths in all age groups per million additionally targeted children. The geographic extension would avert 830 (785, 865) malaria episodes in all age groups per 1000 targeted children under 5, among these 16 (12, 19) severe cases per 1000 targeted children, as well as 137 (107, 162) malaria deaths per million targeted children. The

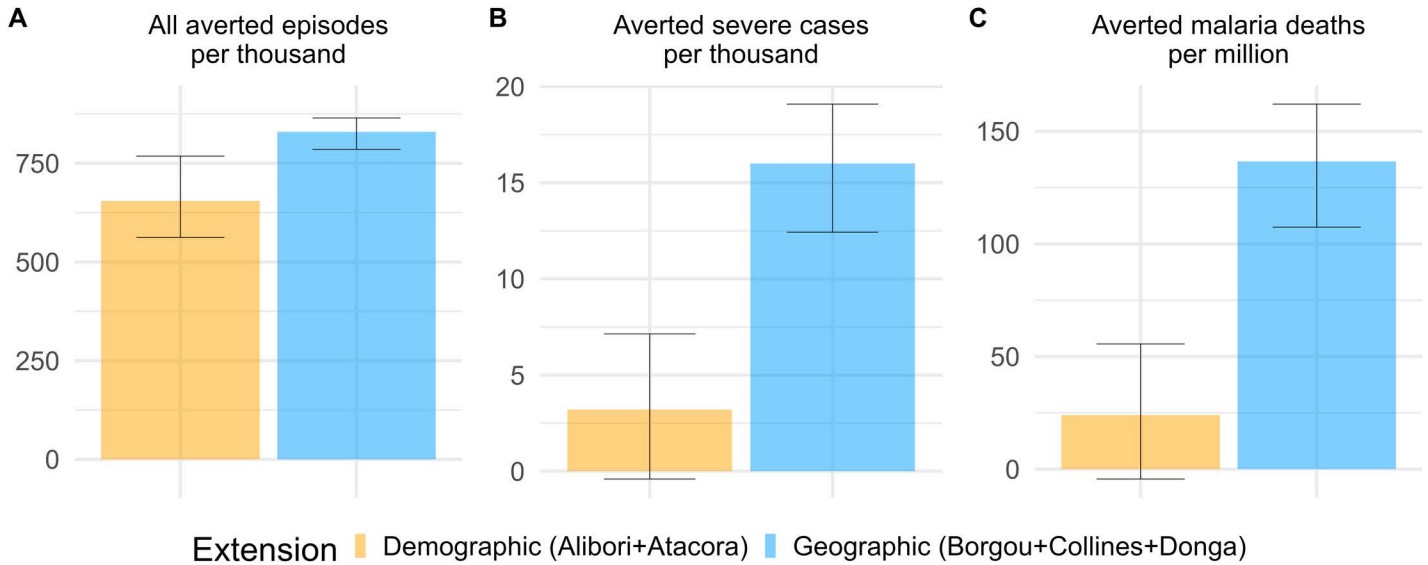

**Fig 6. Cases averted per additionally targeted children between 2024 and 2026.** (A) All malaria episodes in all age groups averted by each SMC extension in eligible zones per thousand additionally targeted children (from 5 to 10 for demographic extension and under 5 for geographic extension). (B) Severe cases averted per thousand additionally targeted children; C: Malaria deaths averted per million additionally targeted children. Error bars represent uncertainty on intensity of transmission as well as model stochasticity, as described in the Methods section.

geographic extension would thus avert on average 1.27 times more episodes, 4.5 times more severe cases, and 5.7 more malaria deaths per targeted child than the demographic extension. Since SMC is recommended to "children belonging to age groups at high risk of severe malaria" [4], the geographic extension appears far preferable as it averts 9 times more severe cases in absolute numbers and 4.5 times more severe cases per targeted children.

### Prioritisation of health zones among geographic extension

When looking at the number of malaria episodes and severe cases averted by the geographic extension of SMC per targeted child by department, no ranking emerged: in Borgou 839 (874, 796) malaria episodes in all age groups could be averted per 1000 children under 5 between 2024 and 2026, among those 17 (13, 20) severe cases per 1000 children, as well as 142 (115, 167) malaria deaths per million children. In Donga 809 (779, 829) malaria episodes, among which 15 (10, 18) severe cases could be averted per 1000 children, as well as 128 (89, 153) malaria deaths per million children. This similarity was also observed at the operational level of health zones (Fig 7).

### Discussion

Our results indicate that the geographic extension would avert at least four times more severe cases and five times more direct malaria deaths per targeted child compared to the demographic extension. This is consistent with the malaria burden distribution in high-endemic settings where children under 5 are most at risk for malaria, with most malaria severe cases and deaths occurring in children under 5 [14,53]. However cases averted by target population appear to be similar within the different zones eligible for the geographic extension.

These results were communicated to the NMCP together with a dashboard to explore different epidemiological indicators, and the NMCP decided to include the geographic extension of SMC in the strategic plan. SMC is administered by community health workers since 2024, and so has first been extended to zones where community health workers have

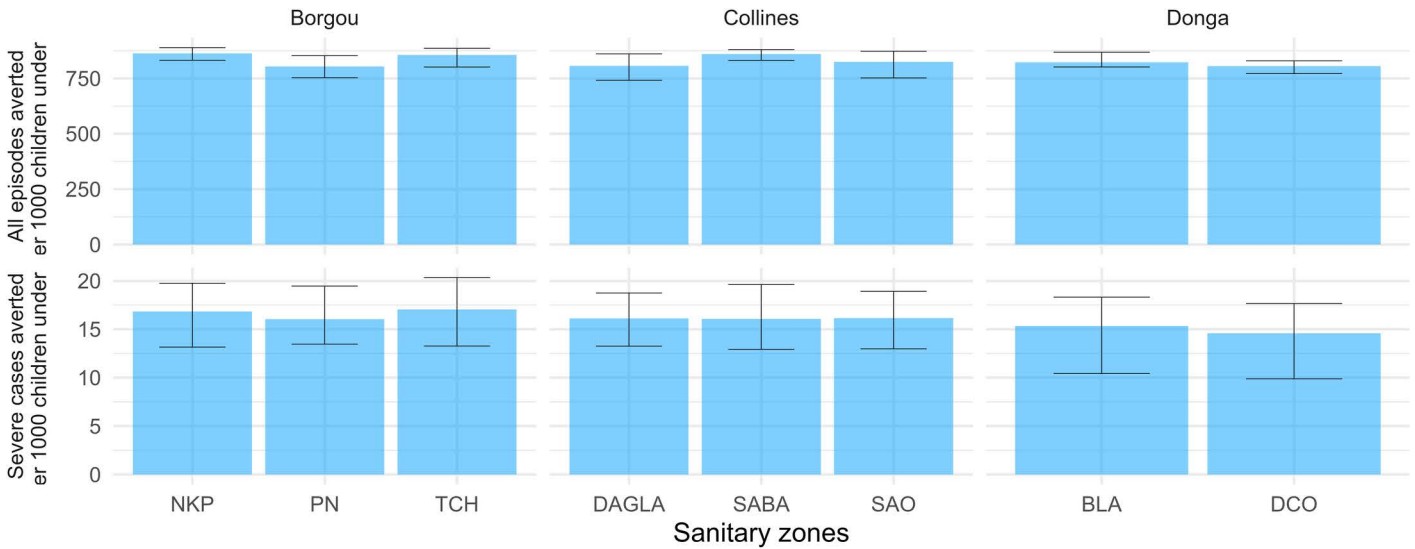

**Fig 7. All malaria episodes and severe malaria cases averted by the geographical extension of SMC per 1000 children under 5 by health zone between 2024 and 2026.** Each acronym stands for one health zone (made of one or several communes): NKP: Nikki-Kalale-Perere; PN: Parakou-N'Dali; TCH: Tchaourou; DAGLA: Dassa-Zoume-Glazoue; SABA: Savalou-Bante; SAO: Save-Ouesse; BLA: Bassila; DCO: Djougou-Copargo-Ouake. Error bars represent uncertainty on intensity of transmission as well as model stochasticity, as described in the Methods section.

already been recruited and trained, in Donga and Collines. Our results are relevant for and specific to Benin in 2024, but a similar approach could be used to support other countries identifying the most impactful implementation of an intervention.

Mathematical modelling can synthetize available evidence on intervention efficacy and extrapolate to specific contexts to support local decision-making. Modelling can account for multiple data sources (seasonality, burden, planned interventions, coverage) to determine which strategy is likely to save most lives. Previous works using mathematical modelling for strategic planning focused on showing the likely impact of already determined combinations of interventions to advocate for funding [23,25]. This analysis shows that country-specific models can also be used to guide the development of strategies by answering to specific questions of interest for decision makers.

The superiority of the geographic extension scenario in northern Benin is most likely due to the over-representation of children under 5 among severe cases in high transmission settings. In OpenMalaria severe malaria episodes can result from high parasitaemia and/or co-occurrence of risk factors, with the latter increasing with EIR and strongly decreasing with age [45]. In all of Benin's departments under consideration for SMC extensions, malaria risk is high: we therefore estimated high EIR values during our calibration (up to 176 infective bites per person per year in Tanguieta, Atacora) and this explains why our results attributed around 90% of severe cases to children under 5. This is in line with several studies which found that at high transmission intensities, young children bear most of the burden of severe cases [12–14]. Since the WHO recommends to target SMC to "children belonging to age groups at high risk of severe malaria" [4], we considered severe cases as one of our main indicators, which is why geographically extending SMC to more children under 5 was found to be more beneficial – but it may not be so in areas with different transmission intensities, or with other interventions already targeting young children. Severe cases and deaths are rare events which are sensitive to stochasticity, so averted severe cases and deaths have large uncertainty bounds which can sometimes contain negatives values without corresponding to real events.

The choice between the two extension scenarios can also be influenced by their anticipated relative cost. From 2019 to 2023 SMC in Benin was delivered as campaigns, with specific staff trained each year, but since 2024 SMC is administered by community health workers. The previous unit costs, which include commodity but also delivery and training

costs, are thus obsolete since the administration of SMC can now rely on pre-existing trained health workers and delivery mechanisms, rather than having to put them in place for each campaign. From the cost drivers identified by Pitt et al. [54], only supply chain and sensitisation would likely cost more in the geographic extension since implementation would happen for the first time, but these represent only 1.5% of the total costs and would only be more expensive during the first year of implementation. Purchase of drugs, however, is the second contributor to costs (27.7%) and from six years of age children need to receive one and a half tablet of both SP and AQ, compared to one tablet from 2 to 5 years included [6]. Although a detailed costing is beyond the scope of this work, the difference in impact is such that the demographic extension would have to cost at least four times less per targeted child to avert the same number of severe cases or deaths per dollar spent. The health zones within the geographic extension were modelled with different seasonalities, history of interventions and transmission risk, but episodes and severe cases averted per targeted child were similar given the level of uncertainty. This means the differences in risk we took into account in our model were not strong enough to drive a differentiated impact given the level of model and data uncertainties. Other considerations, such as the cost of extending to each zone, could have been taken into account to prioritise where to implement SMC first. In the end, the availability of community health workers guided the order of implementation.

This work has nonetheless some limitations. It relies on a model of malaria transmission, which is a simplification of reality. Many parameters of the model were tailored to match local data, which implies that collected data was trusted to reflect reality. EIR was estimated based on prevalence data, and we assumed that all changes in malaria burden were due to interventions, which made the model less flexible to reproduce all past trends perfectly. In practice, climate or socio-economic changes also play a role, but they should influence similarly all considered scenarios and hence not alter our conclusions which rely only on relative comparisons. Given the lack of granularity in the data, some conservative uniformity assumptions were also made: net usage and treatment seeking were assumed uniform across each department, *Anopheles* species composition, and apparition of resistance identical across the entire country. Even though geographical variations in these factors could be relevant, the absence of reliable data prevents us from including such variations in our model: our conclusions thus provide a recommendation on the best strategy given the available evidence, as required for decision-making. There are also fixed model parameters, estimated as part of previous OpenMalaria calibration work: (e.g., species-specific efficacy of nets [36], efficacy of treatments [44,51], and immunity development mechanisms [29,55]), but these refer mainly to the biology of malaria and are unlikely to differ in Benin. The variables with the strongest influence on the local impact of SMC, namely seasonality, other interventions deployed at the same time as SMC, and SMC coverage, were all estimated from Benin data.

We had to calibrate to global estimates as there were no yearly estimates of prevalence in Benin available. The 2011 and 2017 household surveys [38] were used to produce the prevalence estimates, while the 2015 survey [39] was not and thus also not used for calibration. When including this 2015 survey in the comparison of departmental modelled monthly prevalence to survey prevalence, we obtained an adjusted $R^2$ of 0.59. When validating modelled malaria cases, as expected the number of modelled malaria episodes was substantially higher than both the World Malaria Report and routine data, due to OpenMalaria's large threshold for clinical case definition and the double counting of illnesses lasting over 30 days [51].

We reproduced the efficacy of SMC observed in a randomised controlled trial [46], though in real-life settings SMC will likely avert less cases due to less controlled implementation. However, this challenge will probably affect the implementation of geographic and demographic extensions in a similar manner so correcting for this simplification is unlikely to change our results. We assumed SP-AQ had the same efficacy in children under and above 5 years of age, for lack of age-specific data on efficacy. If the drug efficacy varies with age this will in turn affect our results in the same direction (more cases averted in the age group with higher efficacy). However similar odds ratios of mortality and incidence during transmission seasons [8] seem to indicate efficacy is similar in both age groups. Moreover the difference in severe cases averted is so strong that the efficacy in older children would have to be at least four times higher to change our

conclusions, which is unlikely. It is crucial for mathematical models to be able to reproduce the effect sizes of SMC in different ages to guide decisions.

Geographic heterogeneity in future SMC coverage and population structure could not be included for lack of data. We assumed SMC coverage would stay constant through time, space, and age groups. In Senegal a higher coverage in older children was observed by a few percentage points [6], but even if these results were valid for Benin the difference would not be strong enough to change our conclusions. At each round 80% of eligible children were randomly selected to receive SMC, so each child was equally likely to receive the intervention at each round, when in practice some children are systematically missed. In Baba et al. [7], the percentage of children who received four treatments of SMC was higher than the mean monthly coverage to the power of four in all countries but Chad, indicating that the selection of children is not random [25]. However we did not have data on the proportion of children who received each number of rounds, and by randomly selecting 80% of children at each round we obtain on average 41% of children receiving all four rounds, which is a conservative assumption when comparing to the 51.8% children who received all four doses according to the 2021 SMC campaign report. A modelling study [21] found that for another chemoprevention method (Mass Drug Administration) there was little difference in impact between allocating the intervention to the same children or randomly at each round.

The model does not account for the possible emergence of resistance to sulfadoxine-pyrimethamine, but a study using samples collected in 2017 in Benin [56] did not find evidence of widespread resistance in the country. Moreover, modelling evidence [57] indicates that SMC leads to a slow spread of resistance, especially when SMC is administered during four rounds and to children under 5, which happened to be the recommended strategy. The risk of SP resistance developing thus seems minor over the considered time frame.

Population estimates rely on the 2013 census [35], with a 3.51% national growth rate applied to each commune, which was the method recommended by the NMCP. These estimates are outdated: a population count before the 2020 mass net distribution campaign found the population to be 13.6% larger than projected [58], but at the time of analysis results from the new census had not been published. Population growth is also not uniform across departments: between 2002 and 2013, population increased yearly by 2.5% in Collines and 4.6% in Alibori. Since we divided cases averted by the target population, these issues should not influence our results. The proportions of children under 5 (17.4%) and between 5 and 10 (13.7%) are also taken from the 2013 census. They appear not to have changed drastically during the past decade [59].

In Benin, the geographic extension of SMC to southern departments in 2024 would avert at least four times more severe cases per additionally targeted child than its demographic extension, and five times more direct malaria deaths. In zones eligible to the geographic extension SMC was predicted to avert similar number of episodes and severe cases per child under 5. The NMCP adopted the recommendations in their strategy and secured funding for the geographic extension starting in 2024. This country-led collaboration, together with a country-specific modelling approach, led to evidence-based strategic decisions which could be fully funded.

## Supporting information

**S1 Appendix. Figure A1: Population by commune in 2013, 2020 and 2026.** Figure A2: Proportion of population in each age group in Benin according to 2013 census. Figure A3: Monthly incidence by department averaged between 2011 and 2017. Figure A4: ITN use assumptions by department. Figure A5: Type of nets distributed or planned to be distributed in 2020, 2023 and 2026. Figure A6: Assumptions for ingredients and population coverage of IRS campaigns between 2011 and 2021. Figure A7: Case management cascade from access to care measured in 2017–2018 Demographic and Health Survey at the national level. Figure A8: Evolution of access to care (blue) and effective treatment coverage (green) by department. Figure A9: SMC implementation in children under 5 between 2019 and 2023. Figure A10: Calibrated simulations for communes of Alibori. Figure A11: Calibrated simulations for communes of Atacora. Figure A12: Calibrated simulations for communes of Atlantique. Figure A13: Calibrated simulations for communes of Borgou. Figure A14: Calibrated

simulations for communes of Collines. Figure A15: Calibrated simulations for communes of Couffo. Figure A16: Calibrated simulations for communes of Donga. Figure A17: Calibrated simulations for Cotonou, commune of Littoral. Figure A18: Calibrated simulations for communes of Mono. Figure A19: Calibrated simulations for communes of Ouémé. Figure A20: Calibrated simulations for communes of Plateau. Figure A21: Calibrated simulations for communes of Zou. Figure B1: Percentage of reduction of all predicted malaria episodes or severe cases induced by SMC demographic (A) or geographic (B) extension between 2024 and 2026 by age group. Figure B2: Absolute averted malaria episodes, severe cases and deaths by each extension scenario.
(DOCX)

## Acknowledgments

The authors would like to thank Tatiana Alonso Amor, Mar Velarde, Flavia Camponovo, Monica Golumbeanu, and Didier Adjakidje for valuable discussions, George Shireff for past modelling analyses in Benin, and Thomas Smith for proofreading the manuscript. Calculations were performed at sciCORE (http://scicore.unibas.ch/) scientific computing centre at the University of Basel (Basel, Switzerland).

## Author contributions

**Conceptualization:** Jeanne Lemant, Emilie Pothin, Clara Champagne, William Houndjo, Julien Aïssan, Rock Aïkpon, Sakariahou Kpanou, Cyriaque Affoukou.

**Data curation:** Jeanne Lemant, William Houndjo, Julien Aïssan, Camille Houetohossou, Rock Aïkpon.

**Formal analysis:** Jeanne Lemant, Emilie Pothin, Clara Champagne, William Houndjo, Julien Aïssan, Rock Aïkpon, Sakariahou Kpanou, Cyriaque Affoukou.

**Funding acquisition:** Emilie Pothin.

**Investigation:** Jeanne Lemant, William Houndjo, Julien Aïssan, Rock Aïkpon, Sakariahou Kpanou, Cyriaque Affoukou.

**Methodology:** Jeanne Lemant, Emilie Pothin, Clara Champagne.

**Project administration:** Jeanne Lemant, Emilie Pothin.

**Resources:** Jeanne Lemant, Emilie Pothin, William Houndjo, Julien Aïssan, Camille Houetohossou, Rock Aïkpon, Sakariahou Kpanou, Cyriaque Affoukou.

**Software:** Jeanne Lemant, Roland Goers.

**Supervision:** Emilie Pothin, Clara Champagne.

**Validation:** Jeanne Lemant.

**Visualization:** Jeanne Lemant.

**Writing – original draft:** Jeanne Lemant.

**Writing – review & editing:** Jeanne Lemant, Emilie Pothin, Clara Champagne, William Houndjo, Julien Aïssan, Camille Houetohossou, Rock Aïkpon, Sakariahou Kpanou, Roland Goers, Cyriaque Affoukou.

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
