## [Decision Letter · Decision Letter 0]

29 Oct 2024

PGPH-D-24-02000

Supporting evidence-based decisions about the geographic and demographic extensions of seasonal malaria chemoprevention in Benin: a modelling study

Dear Dr. Pothin,

Thank you for submitting your manuscript to PLOS Global Public Health. After careful consideration, we feel that it has merit but does not fully meet PLOS Global Public Health’s publication criteria as it currently stands. Therefore, we invite you to submit a revised version of the manuscript that addresses the points raised during the review process.

We look forward to receiving your revised manuscript.

Kind regards,

Mathieu Nacher

Academic Editor

Journal Requirements:

1. Please note that PLOS ONE has specific guidelines on code sharing for submissions in which author-generated code underpins the findings in the manuscript. In these cases, we expect all author-generated code to be made available without restrictions upon publication of the work. Please review our guidelines at https://journals.plos.org/plosone/s/materials-and-software-sharing#loc-sharing-code and ensure that your code is shared in a way that follows best practice and facilitates reproducibility and reuse.

Additional Editor Comments (if provided):

Reviewers' comments:

Reviewer's Responses to Questions

**Comments to the Author**

1. Does this manuscript meet PLOS Global Public Health’s publication criteria?

Reviewer #1: Partly

Reviewer #2: Partly

2. Has the statistical analysis been performed appropriately and rigorously?

Reviewer #1: N/A

Reviewer #2: No

3. Have the authors made all data underlying the findings in their manuscript fully available (please refer to the Data Availability Statement at the start of the manuscript PDF file)?

Reviewer #1: No

Reviewer #2: No

4. Is the manuscript presented in an intelligible fashion and written in standard English?

Reviewer #1: Yes

Reviewer #2: Yes

Reviewer #1: This a paper modelling seasonal Malaria Chemoprevention to compare treating 0-5 children in previously untreated areas vs extending SMC to 5-10 in previously treated areas.

The authors conclude that geographical extension is likely to prevent more cases than demographic extension.

The research appears to be sound, they are using a "OpenMalaria" model, but I could not see the code. I believe this would be important

if others want to apply the same kind of analysis.

The article focuses on a particular situation (Benin) and is essentially a report geared at the health authorities.

It is argued that this is relevant since the comparison

builds on many parameters and could change depending on the particular environment.

However, this makes it difficult to understand why the conclusion leaned towards a particular choice, and

then what could be extrapolated to other locations, other populations.

At the same time, the study is not really a "how to" aiming at making he same computation in other situations.

There are a number of parameters that are fixed to default values due to lack of data.

As it is a pure modelling study, the paper could be made more general by trying to illustrate what is the generality

that targeting 0-5 is better than adding 5-10. Identifying what conditions make it so.

other points :

ABSTRACT

'likely more cost-effective' should be removed. You did not perform cost effectiveness analysis.

Are these the same children receiving SMC during the season or is it sampled on each of the 4 rounds? does it change?

It is difficult to follow what is necessary to establish a scenario (nets, treatment, efficacy).

A table listing parameters and sources would be welcome.

No data is presented in the main text to illustrate that the model captures the underlying situation satisfactorily.

Fig A.9-A.13 does not show a particular good fit to the data: the average is indeed calibrated, but the model does not seem to reproduce the observed trends. How did the authors decide that the model provided a satisfactory fit to the data? Would the lack of fit in trends alter the conclusions ?

In the priorisation study, what could be the differences leading to a change in impact? Aren't those territories modelled all the same,

using the same model ?

Reviewer #2: This work presents a study estimating the impact of interventions planned to control malaria transmission in Benin using a mathematical modeling approach. The problem is undoubtedly relevant, and the approach is valid, despite certain limitations acknowledged by the authors. However, in my view, the study does not adequately describe the methodology. Given a baseline, which is calibrated in the model, the authors could have provided the baseline numbers.

I understand that the effect of planned interventions is estimated using parameters such as ITN coverage and drug treatment. However, it is unclear whether the planned interventions are already in place or still in development. My primary concern revolves around the concept of a baseline. The study compares the impact of additional layers of interventions with the planned interventions. Another issue is that the comparison involves uncertainties on both sides—the effect of the planned interventions and the effect of the additional interventions.

If this interpretation is correct, the methodology section should provide a clearer explanation of how these comparisons are made. Furthermore, in the results, the overall epidemiological profile and baseline should be presented, leaving a reader unfamiliar with malaria in Benin without a clear understanding of the number of cases involved.

Also, an analysis of cost effectiveness was not done, instead the impact was estimated via numbers of avoided cases. I suggest refraining from taking results on the basis of cost-effectiveness.

Additional comments:

Introduction

The statement about the goal says that the goal is to verify which extensions on interventions are cost-effective, however a rigorous cost-effectiveness analysis was not performed.

Methodology

The methodology should include a description of the study area—in this case, the Benin population—and how cases have been reported over the past years (including sources of data).

How was the EIR in 2000 obtained? It is unclear whether it comes from a previous publication or was estimated through a specific methodology. The authors mention that lower and upper bound estimates for EIR were used to account for uncertainty, but how were these estimates determined?

Why were 10 infections per 1,000 people considered for imported cases?

What is the evaluation period? Are all simulation scenarios based on a two-year period? This was not clear. I suggest providing a table or a descriptive list of the scenarios.

Table 1 contains figures—perhaps it would be better as a panel of figures.

The number of newly targeted children is used a proxy for cost-effectiveness. A truly cost-effectiveness analysis would involve the financial costs. I understand that the study quantifies the impact and the number of newly targeted children is a measure of the effort of the additional interventions.

Results

I believe the baseline is missing from the manuscript. Possibly a figure showing the time series of cases up to 2023, or a table with demographic information (such as areas, cases, and severe cases) should be included.

Additionally, a profile of the population exposed to malaria could be helpful, possibly presented in a table. The supporting info material has additional information that could bring this notion.

For Figure 2 (and subsequent figures), what is the time interval?

Figure 4 introduces terms that were not previously defined, such as NKP and DAGLA. The caption, “Error bars represent uncertainty in transmission intensity as well as model stochasticity,” suggests that uncertainty arises from both factors, possibly implying they are independent. However, I would argue that stochasticity is used in the model to describe this uncertainty. Therefore, I suggest rephrasing this to avoid misleading interpretations.

Discussion

In the paragraph on limitations in the Discussion, the authors should explain why these limitations can be accepted—whether the biases are expected to be minimal or if a more conservative approach is warranted.

I believe the issue of drug resistance should be addressed. This is important when expanding interventions as proposed here.

**Do you want your identity to be public for this peer review?** For information about this choice, including consent withdrawal, please see our Privacy Policy

Reviewer #1: No

Reviewer #2: No

---

## [Decision Letter · Decision Letter 1]

26 Feb 2025

PGPH-D-24-02000R1

Supporting evidence-based decisions about the geographic and demographic extensions of seasonal malaria chemoprevention in Benin: a modelling study

Dear Dr. Pothin,

Thank you for submitting your manuscript to PLOS Global Public Health. After careful consideration, we feel that it has merit but does not fully meet PLOS Global Public Health’s publication criteria as it currently stands. Therefore, we invite you to submit a revised version of the manuscript that addresses the points raised during the review process.

We look forward to receiving your revised manuscript.

Kind regards,

Mathieu Nacher

Academic Editor

Journal Requirements:

Additional Editor Comments (if provided):

Reviewers' comments:

Reviewer's Responses to Questions

**Comments to the Author**

Reviewer #2: (No Response)

publication criteria?

Reviewer #2: Yes

3. Has the statistical analysis been performed appropriately and rigorously?

Reviewer #2: Yes

4. Have the authors made all data underlying the findings in their manuscript fully available (please refer to the Data Availability Statement at the start of the manuscript PDF file)?

Reviewer #2: Yes

5. Is the manuscript presented in an intelligible fashion and written in standard English?

Reviewer #2: Yes

Reviewer #2: The manuscript improved significantly with more information on the methodology, updates on the results, and several other aspects. I believe now it lacks conveying in the abstract the notion that the geographical extension is advantageous because of the effect on children ages 0-5.

I also recommend a revision on some points that appear in the Results, however they suit better in the Discussion.

Other comments below:

- The abstract has to reflect the point that due to the frequency of severe cases in ages 0-5, the geographical extension had better performance.

- Line 47: provide the location for the number of cases

- Line 171: remove “by far”

- Line 204: correct “ Once an individual seeks treatment, they do not necessarily”

- Be more clear in the sentence: “Because the standard deviation associated with each observation point is accounted for, the pseudo-likelihood assigns more importance to observations with lower uncertainty.” I think it requires a better understanding both in terms of “standard deviation associated with an observation point” and “more importance to lower uncertainty”.

- It is unclear why simulations were initially run with 50,000 of population size if later the actual population sizes were used, which makes sense.

- Line 316:  the number of averted cases would be the difference between the estimate obtained with an extension and the number given the planned interventions.  Hence, should not be I_extension minus I_planned?

- Line 356:  MIS and DHS were not defined.  Also, the consideration in this sentence should be in the Discussion.

- Line 387: I believe this consideration is for the Discussion.

- Line 411:  the intervals require a different dash character to distinguish negative numbers. Also, the Discussion would benefit from the interpretation of these negative numbers.

- Line 447: again better fit in the Discussion.

- Line 495-496: why unit costs obsolete?

- Line 504 - 505: I recommend some caution is required as the sentence provides premature statements.

- Line 602:  I also recommend refraining from talking about “return on investment.”

**Do you want your identity to be public for this peer review?** For information about this choice, including consent withdrawal, please see our Privacy Policy

Reviewer #2: No

---

## [Editor Report · Decision Letter 2]

26 Mar 2025

Supporting evidence-based decisions about the geographic and demographic extensions of seasonal malaria chemoprevention in Benin: a modelling study

PGPH-D-24-02000R2

Dear Dr. Pothin,

We are pleased to inform you that your manuscript 'Supporting evidence-based decisions about the geographic and demographic extensions of seasonal malaria chemoprevention in Benin: a modelling study' has been provisionally accepted for publication in PLOS Global Public Health.

Best regards,

Mathieu Nacher

Academic Editor